# Reestablishment of spermatogenesis after more than 20 years of cryopreservation of rat spermatogonial stem cells reveals an important impact in differentiation capacity

**Eoin C. Whelan**[1], **Fan Yang**[1,2], **Mary R. Avarbock**[1], **Megan C. Sullivan**[3], **Daniel P. Beiting**[3], **Ralph L. Brinster**[1] *

**1** Department of Biomedical Sciences, School of Veterinary Medicine, University of Pennsylvania, Philadelphia, Pennsylvania, United States of America, **2** Department of Histology and Embryology, Medical College, Yangzhou University, Yangzhou, China, **3** Department of Pathobiology, School of Veterinary Medicine, University of Pennsylvania, Philadelphia, Pennsylvania, United States of America

* brinster@vet.upenn.edu

## Abstract

Treatment of cancer in children is increasingly successful but leaves many prepubertal boys suffering from infertility or subfertility later in life. A current strategy to preserve fertility in these boys is to cryopreserve a testicular biopsy prior to treatment with the expectation of future technologies allowing for the reintroduction of stem cells and restoration of spermatogenesis. Spermatogonial stem cells (SSCs) form the basis of male reproduction, differentiating into all germ cell types, including mature spermatozoa and can regenerate spermatogenesis following transplantation into an infertile testis. Here, we demonstrate that rat SSCs frozen for more than 20 years can be transplanted into recipient mice and produce all differentiating germ cell types. However, compared with freshly isolated cells or those frozen for a short period of time, long-frozen cells do not colonize efficiently and showed reduced production of spermatids. Single-cell RNA sequencing revealed similar profiles of gene expression changes between short- and long-frozen cells as compared with fresh immediately after thawing. Conversely, following transplantation, long-frozen samples showed enhanced stem cell signaling in the undifferentiated spermatogonia compartment, consistent with self-renewal and a lack of differentiation. In addition, long-frozen samples showed fewer round spermatids with detectable protamine expression, suggesting a partial block of spermatogenesis after meiosis resulting in a lack of elongating spermatids. These findings strongly suggest that prolonged cryopreservation can impact the success of transplantation to produce spermatogenesis, which may not be revealed by analysis of the cells immediately after thawing. Our analysis uncovered persistent effects of long-term freezing not found in other cryopreservation studies that lacked functional regeneration of the tissue and this phenomenon must be accounted for any future therapeutic application.

**Data Availability Statement:** All raw sequence files and processed files have been submitted to NCBI GEO repository GSE182438.

**Funding:** This work was funded by Robert J. Kleberg, Jr and Helen C. Kleberg Foundation (RLB). https://www.klebergfoundation.org/grant-guidelines/medical-research/ The funders had no role in study design, data collection and analysis, decision to publish, or preparation of the manuscript.

**Competing interests:** The authors have declared that no competing interests exist.

**Abbreviations:** DEG, differentially expressed gene; MACS, magnetic-activated cell sorting; PCA, principal component analysis; scRNA-seq, single-cell RNA sequencing; SNP, single nucleotide polymorphism; SSC, spermatogonial stem cell; UMAP, uniform manifold approximation and projection; UMI, unique molecular identifier.

## Introduction

The rat has long been an important model of mammalian spermatogenesis, and the histological process of rat male germ cell differentiation is well understood [1]. An overview of the differentiation program is shown in Fig 1A. Spermatogonial stem cells (SSCs) form the basis for lifelong sperm production as they can both self-renew and differentiate into all germ cell types. SSCs have the potential to regenerate spermatogenesis when transplanted into a recipient testis that lacks germ cells and therefore have therapeutic applications for preserving or modifying the male germ line. SSCs undergo proliferative mitotic divisions, first as undifferentiated progenitor spermatogonia and then as differentiating spermatogonia, from type A to intermediate to type B, resulting in an approximately 500-fold increase in cell number during this process. Type B spermatogonia then divide and enter meiosis as preleptotene spermatocytes, becoming leptotene and then zygotene spermatocytes. In this work, we define early spermatocytes as those up to the zygotene stage. The next, and longest, stage of meiosis is the pachytene stage where spermatocytes complete the complicated process of chromatin condensation and crossing over. In rats, this takes approximately 12 days. The diplotene stage and meiotic divisions into secondary spermatocytes are comparatively fast, only 18 and 15 hours, respectively. Consequently, these cell types are difficult to see in histological staging [1], and grouped with pachytene spermatocytes we designated these as late spermatocytes. Finally, following meiosis, spermiogenesis begins. This is a lengthy process of extreme changes as cells become round and then elongating spermatids. Complex chromatin remodeling takes place, first replacing chromatin with spermatid-specific histones, then with transition proteins and finally with protamines. These changes are associated with dramatic cell morphological changes, resulting in elongating spermatids and finally functional sperm. Theoretically, 4,096 spermatids could be generated from one stem cell but many are lost throughout the process. Nevertheless, cells from the later stages of spermatogenesis make up the majority of germ cells in the testis.

Improvements in cancer therapeutics have greatly improved survival rates in children. Survival in pediatric patients has increased from 58% 5-year survival in 1975 to 1977 to 83% in 2001 to 2007 [2]. One serious side effect of these therapies is often reduced fertility or infertility [3]. Sperm can be banked for postpubertal males but not in children. Fertility could potentially be preserved by taking a testicular biopsy, cryopreserving SSCs either in situ as a piece of testis or as dissociated cells to be expanded later [4,5]. There is an urgent need for fertility preservation in prepubertal patients, which is already being performed clinically in various locations around the world to extract and freeze testicular biopsies prior to gonadotoxic therapies in boys [6–8], despite a lack of existing technologies to make use of such cryopreserved tissues [9]. In order to translate into therapies, this procedure relies on the ability to recover SSCs from frozen samples and transplant them into a recipient host. Producing sufficient cells would require culture of human SSCs, which at this time continues to be extremely challenging [10]. Recently, an alternative method has been developed, where pieces of macaque testis have been frozen and thawed and after reengraftment, sperm have been successfully generated and used to produce offspring [11], suggesting that a clinical application may be in sight.

Although this prepubertal patient population could benefit from long-term preservation of SSCs, there is limited data on whether extended freezing of SSCs is safe and effective for reconstitution of spermatogenesis. Long-term cryopreservation of SSCs has been demonstrated in mice [12], but whether cryopreservation has any negative consequences in germ cells after transplantation is unknown. In order to address these questions, we took advantage of rat SSC cells cryopreserved for more than 23 years in our laboratory alongside a continuously maintained colony of the same rat line. Single-cell RNA sequencing (scRNA-seq) afforded us the tools to extract transcriptomic information from individual SSCs in dissociated mixtures of

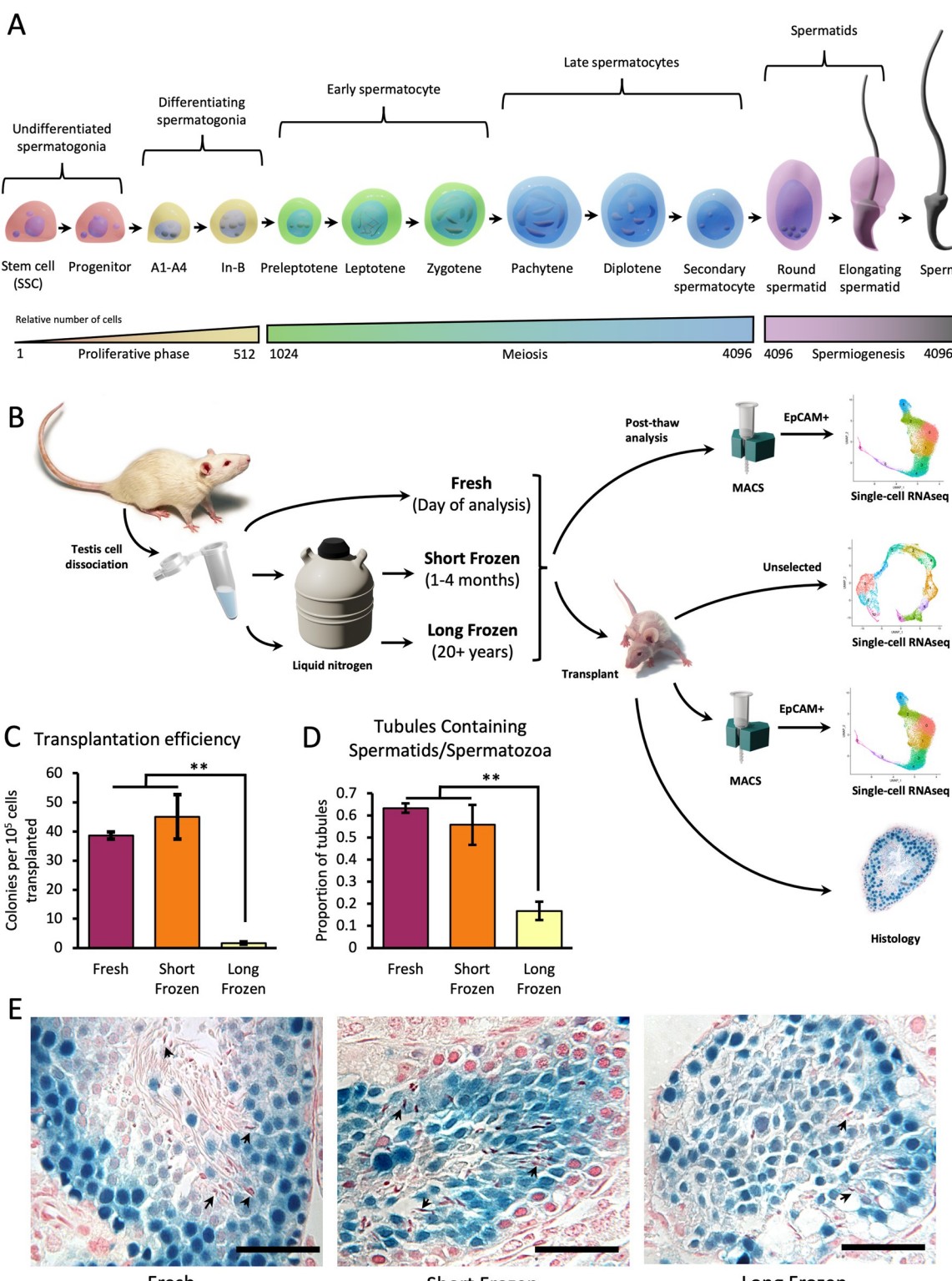

**Fig 1. Transplantation of cryopreserved rat germ cells reconstitute spermatogenesis in the mouse. (A)** Summary of spermatogenesis. Stem cells are the founders of spermatogenesis and differentiate into proliferating progenitor spermatogonia. Differentiating spermatogonia are comprised of A1 to A4 spermatogonia, intermediate (In) and B spermatogonia. Preleptotene, leptotene, and zygotene are grouped as early spermatocytes, while pachytene, diplotene, and secondary spermatocytes are classified as late spermatocytes. Finally, round and elongating spermatids precede fully formed sperm. Ideal numbers of cells are shown given the number of cell divisions from a

single stem cell; however, in actuality, numbers are lower [1]. **(B)** Overview of experimental design. Single cells from digested testes were frozen in 10% DMSO for 20+ years ("long-frozen" cells). This process was repeated for cells frozen for 1–4 months ("short-frozen" cells). Together with fresh cells that have not been frozen, samples were enriched for EpCAM+ cells and either analyzed directly by scRNA-seq or transplanted into a recipient J:Nu mouse. After 3–4 months, transplanted testes were extracted and stained for LacZ colonies or encapsulated for scRNA-seq (both unselected and EpCAM+ for each sample). For each outcome, at least 3 biological replicates were used. **(C)** Transplantation efficiency of cells following freezing treatment. Error bars denote SEM. Significance assessed by ANOVA. $^{**}$ $p < 0.01$. Individual observations recorded in S4 Data. **(D)** LacZ+ tubules scored for presence of elongating spermatid or spermatozoa heads and the fraction reported for each biological replicate (281 tubules scored, $n = 3$). Error bars denote SEM. Significance assessed by ANOVA. $^{**}$ $p < 0.01$. Individual observations recorded in S4 Data. **(E)** Histological sections of representative tubules showing visible spermatids (arrowheads). Scale bar = 50 μm. MACS, magnetic-activated cell sorting; scRNA-seq, single-cell RNA sequencing; SSC, spermatogonial stem cell.

testicular cells as well as to address comparisons between treatment groups. Single-cell analyses of murine [13–19] and human germ cells [14,20–23], as well as those from macaque [24], have provided novel insights into the process of germ cell differentiation. The rat is an important and long-established animal model for germ cell biology, but single-cell transcriptomic analysis of rat germ cell development is lacking. Rat SSCs can be transplanted into a mouse recipient and successfully regenerate spermatogenesis [25], with rat spermatogenesis proceeding according to the rat seminiferous cycle timing [26]. In addition to therapeutic preservation of testicular tissues from prepubertal cancer patients, cryopreservation of SSCs has applications in conservation biology [27], and in long-term preservation of germ cells from valuable agricultural animals [28]. In this study, we report single-cell analysis of rat spermatogenesis and the effect of long-term cryopreservation on both postthaw SSCs and spermatogenesis after transplantation into a mouse host.

## Results

### Transplantation of fresh, short-frozen, and long-frozen germ cells

The goal of this study was to investigate the effect of cryopreservation over long time periods on the ability of rat SSCs to successfully regenerate spermatogenesis. We designed an experiment to take advantage of stocks of Sprague-Dawley *Rattus norvegicus* testis cells that were dissociated and cryopreserved over 23 years before the start of this study (which in this paper we refer to as "long-frozen" cells). We repeated the freezing process and stored the samples in liquid nitrogen for between 1 and 4 months ("short-frozen" cells). In addition to these, we used cells from testes taken on the day of analysis from a colony of the same inbred rats maintained continuously in our laboratory throughout that time ("fresh" samples). Dissociation method and freezing protocols were the same as used 23+ years ago.

To assess the potential for cryopreserved stem cells to colonize empty niches and regenerate spermatogenesis, fresh, short-frozen, and long-frozen samples were thawed and used for selection of EpCAM+ cells to enrich for SSCs before being transplanted into busulfan-treated nude mice (Fig 1B). Busulfan treatment removes endogenous mouse germ cells, leaving empty niches available for colonization by rat germ cells [29,30]. All treatments produced colonies of spermatogenesis, but long-frozen cells showed dramatically and significantly lower numbers of colonies formed per $10^5$ cells transplanted than either short-frozen or fresh cells (Fig 1C, $n = 3$). Fresh cells from this study showed similar colony forming ability to those from studies approximately 20 years earlier (S1A Fig). Histological analysis of the transplanted colonies showed the full spectrum of rat germ cells at different stages including elongated spermatids in all treatments (Fig 1E). While the testis transplanted with long-frozen samples contained occasional sperm, indicating completion of spermatogenesis, properly organized seminiferous tubules with a visible lumen were not observed and the proportion of tubules displaying elongating spermatid or spermatozoa heads was significantly lower (Fig 1D).

## Clustering and identification of rat germ cell types

Given these observed differences between treatments, we selected scRNA-seq as the frozen samples constituted a mixed population of cells, and this technology allowed us to compare the transcriptomic profiles of individual cell types [31]. As the rat testis transcriptome has not been investigated using scRNA-seq, we first built a single-cell resolution map of rat spermatogenesis. Single-cell suspensions from testes were either directly used for scRNA-seq or were first enriched for EpCAM-positive cells via magnetic-activated cell sorting (MACS), selecting for cells in the earlier stages of spermatogenesis that are otherwise greatly outnumbered by post-meiotic cells in the adult rodent testis [32]. After filtering out low-quality cells and cells with high percentage (>20%) of mitochondrial reads, all replicates and treatments of unselected cells were integrated together, clustered and plotted as a uniform manifold approximation and projection (UMAP) [33]. The same process was performed for EpCAM+ cells (S1B and S1E Fig). Clusters expressing somatic cell markers were removed (S2A and S2B Fig), and the remaining cells expressing germ cell markers were reclustered in an unsupervised manner (Fig 2A). Clusters showed distinct transcriptomic profiles using key germ cell genes in unselected and EpCAM+ samples, regardless of freezing treatments (Fig 2B). Pseudotime trajectory of both unselected and EpCAM+ germ cell populations showed a largely linear progression from *Etv5*-expressing undifferentiated spermatogonia at one end to spermatid cells that expressed over 200-fold higher *Prm1* and *Tnp1* levels and 7-fold lower levels of *Etv5* (Fig 2C). Metrics of genes/cell, unique molecular identifier (UMI)/cell, and mitochondrial percentage are shown in S3 Fig.

Clusters were ordered along pseudotime and showed highly divergent identities when aligned against known markers of germ cell progression (Fig 3A, S4A and S4B Fig). Unselected samples showed proportionately many more cells toward the later stages of spermatogenesis consistent with amplifying cell divisions during spermatogenesis (Fig 1A). EpCAM+ samples provided a higher resolution of the early stages of spermatogenesis, including SSCs. Key stem cell–associated genes such as *Etv5* and *Ret* were up-regulated in both unselected and EpCAM+ undifferentiated spermatogonial populations, but only in the EpCAM+ samples could progenitors be distinguished from SSCs by a cluster expressing *Sall4*, *Uchl1*, and *Crabp1*. Differentiating spermatogonia are marked early by *Kit*, a gene essential for spermatogonial differentiation. Entrance into meiosis was indicated by *Stra8* and *Mei1* marking preleptotene cells. Leptotene/zygotene cells expressed *Sycp3* and *Tex101*. Pachytene spermatids were clearly marked by *Piwil1*. Round spermatids showed *Acrv1* and *Catsper3* expression. As the round spermatids matured, they expressed the transition proteins *Tnp1* and *Tnp2* and protamines *Prm1* and *Prm2*, which are up-regulated throughout the transformation into elongating spermatids (Fig 3A). Ingenuity pathway analysis of differentially expressed gene (DEG) sets for each cell type showed a distinct set of pathways (Fig 3B, S1 Data). Stem cells showed strong up-regulation of GDNF, EGF and JAK/Stat signaling. Progenitors and differentiating spermatogonia showed interactions with Sertoli cells or gap junction signaling along with fatty acid biosynthesis, which can be seen as fatty acid activation in spermatocytes, but notably have entirely lost significant stem cell pathway signaling such as GDNF signaling. Cholesterol and pyrimidine biosynthesis is observed in round spermatids, and, finally, nucleotide excision repair and transcriptional repression were enriched in elongating spermatids (Fig 3B). These cell type identities based on marker genes and pathways were applied to all cells for unselected and EpCAM+ cell populations in all treatments (Fig 3C and 3D). In summary, our scRNA-seq analyses delineated rat germ cell types distinctly at the single cell level.

## Effects of cryopreservation on the transcriptome of germ cells after thawing

We analyzed EpCAM-selected populations split between fresh, short-frozen, and long-frozen treatments immediately postthaw. Each of these treatments clustered together and contained

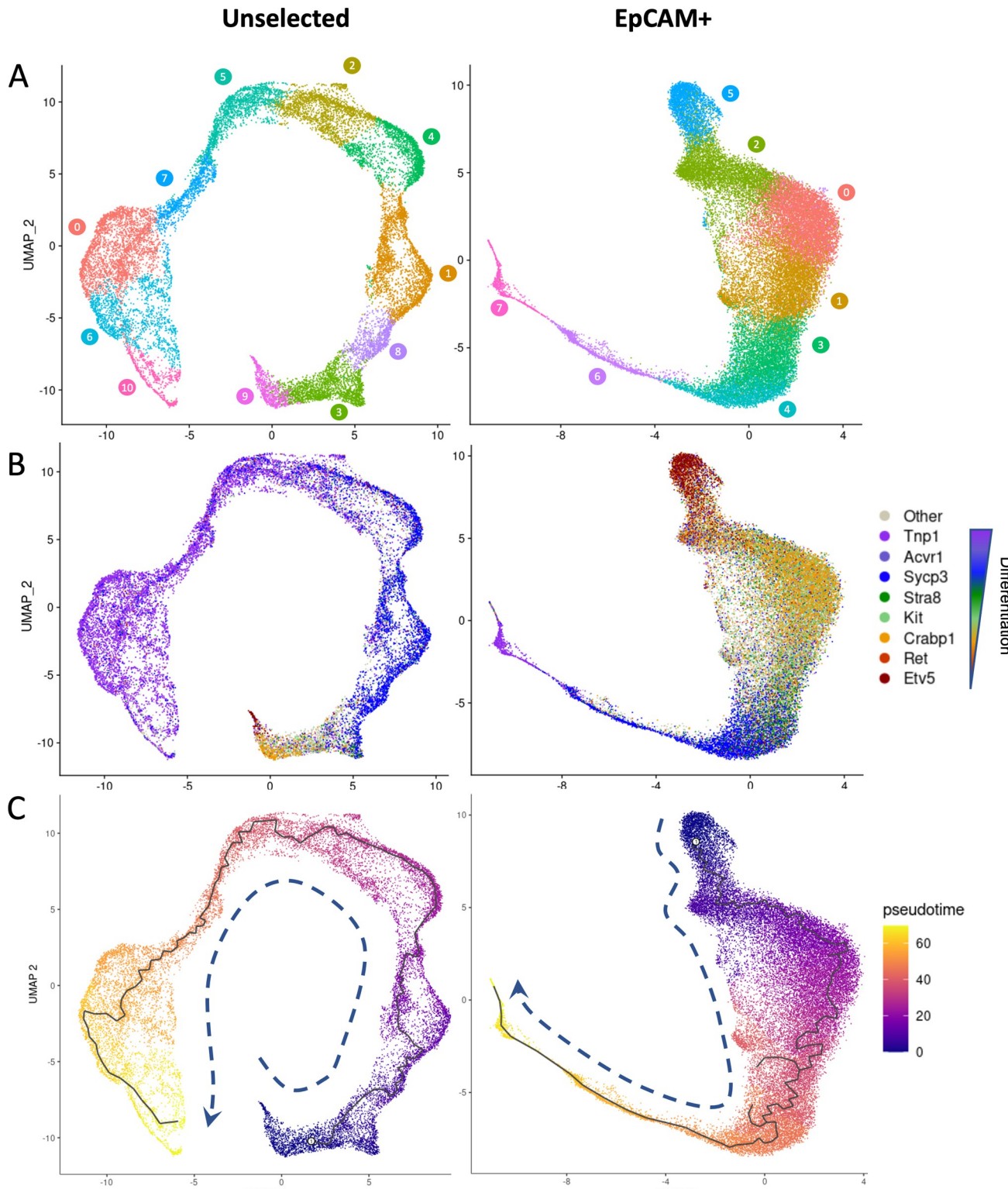

**Fig 2. Key germ cell markers delineate transcriptional states in unsupervised clustering. (A)** Unsupervised clustering of all germ cells, without selection (left) or selected for the surface marker EpCAM (right). **(B)** Distribution of select genes (each cell is colored by gene above a threshold of 1 normalized count per million in order listed) in unselected and EpCAM+ germ cells. **(C)** Pseudotime ordering of unselected and EpCAM+ germ cells. All underlying data deposited in NCBI GEO repository (GSE182438). UMAP, uniform manifold approximation and projection.

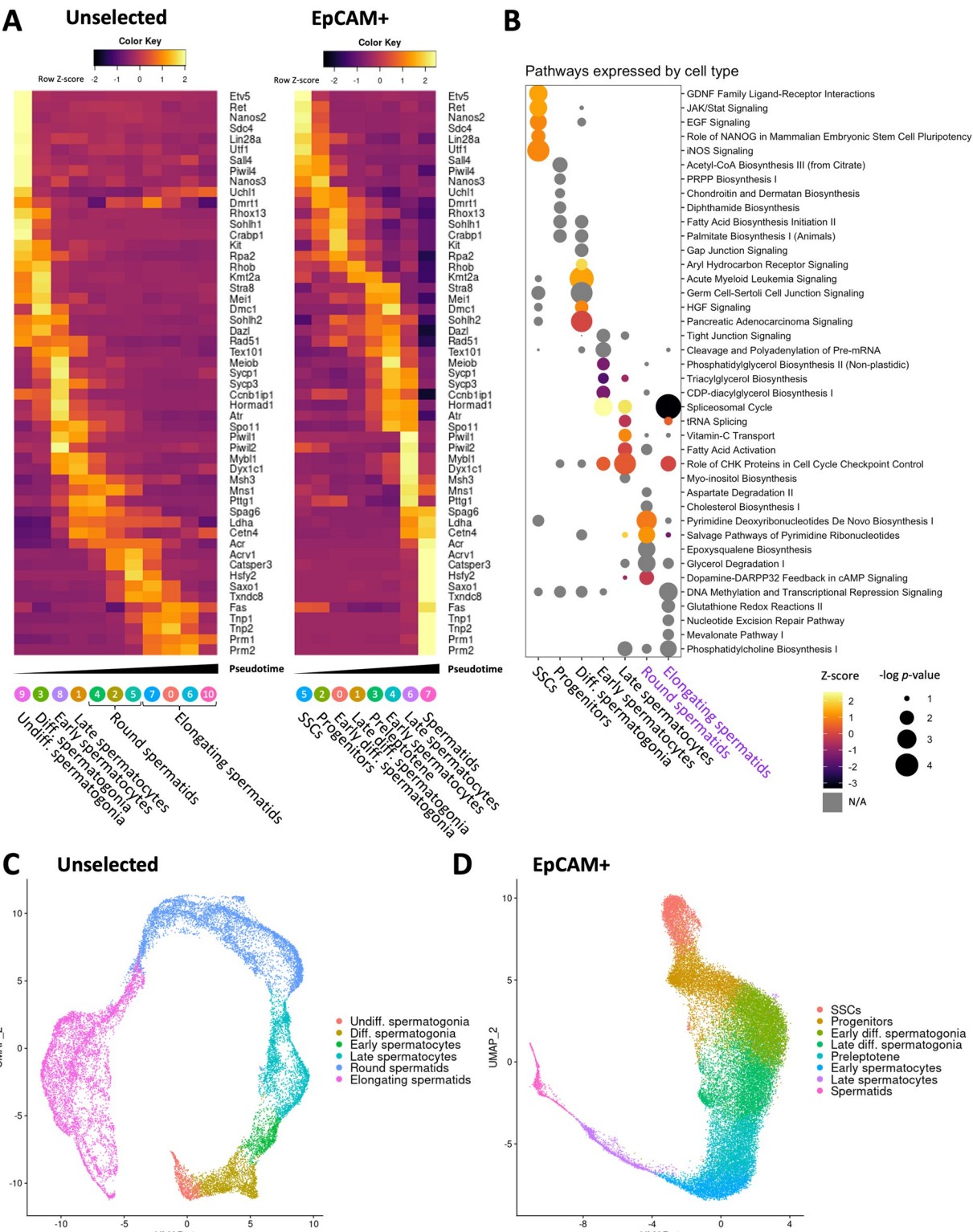

**Fig 3. Rat germ cell identity assignments.** (A) Heatmaps showing selected gene expression of aggregate normalized counts per cell type in unselected and EpCAM+ cell types. Cell type assignments were applied to unsupervised clusters generated in Fig 2 based on expression of marker genes corresponding to known cell types in Fig 1A. Underlying data shown in S4 Data. **(B)** Ingenuity pathway analysis of selected pathways by cell type in germ cells. Shown are EpCAM-derived clusters for early germ cell states (black text) and unselected for postmeiotic cells (purple text). **(C)** Unselected germ cell cluster assignments. **(D)** EpCAM+ germ cell cluster assignments. All underlying data deposited in NCBI GEO repository (GSE182438). SSC, spermatogonial stem cell; UMAP, uniform manifold approximation and projection.

all germ cell types (Fig 4A). We focused on the effect of freezing on SSCs, as these cells would form the basis for regeneration of spermatogenesis in any therapeutic application. To assess overall transcriptomic differences in stem cells between treatments, the counts for each replicate were pseudobulked (i.e., an average expression value for all cells in the SSC cluster) and used for principal component analysis (PCA; Fig 4B). Each treatment had at least 3 independent biological replicates (S5A and S5B Fig). There was no distinct pattern in PCA among short- and long-frozen samples, but fresh showed significant difference in comparison with the frozen samples (PERMANOVA $p$ = 0.008 with 1,000 permutations, aggregating short-frozen and long-frozen as a single treatment and compared with fresh). Single nucleotide polymorphism (SNP) and indel PCA analysis of the mRNA alignments to the rat genome did not detect any patterns of mutations corresponding to treatments (S5C Fig). When looking at the number of genes that are significantly different, the freezing samples are far more similar to each other than to the fresh, showing only 13 genes significantly different. While little significant difference was observed between short- and long-frozen treatments, long-frozen samples showed more significant transcriptional differences compared to fresh than the comparison of short-frozen to fresh (Fig 4C). These results can be reconciled by short- and long-frozen showing similar changes in direction, but more extreme magnitude in the case of long-frozen. Consistent with these data, pathway analysis showed that when a given pathway was affected by freezing, it was often more significantly perturbed in cells that were long-frozen, compared to short-frozen (Fig 4D). For example, pathways involved in cell stress and protein synthesis were up-regulated in both frozen samples. After thawing, frozen samples showed distinct transcriptional changes to fresh but little significant difference between freezing treatments.

## Single-cell analysis of transplanted spermatogenesis

Single-cell suspensions were prepared from transplanted testes by digestion. Both unselected and EpCAM+ cell fractions were encapsulated for scRNA-seq. As the recipients were mice, it was important to distinguish between rat spermatogenesis and any endogenous mouse spermatogenesis. Busulfan treatment efficiently removes endogenous mouse germ cells, but a small proportion of tubules escape ablation. Identification of rat cells was achieved by aligning samples against both mouse and rat reference genomes and for each cell, assigning an identity based on which reference generated more UMI counts above a minimum threshold of 5% more than the other genome, otherwise cells were marked as "unknown" (S1C and S1F Fig). When tested on pure mouse and pure rat samples, this method proved to be at least 99.9% accurate in assigning species identity to cells (S1D and S1G Fig). Only cells identified as rat were included in reclustered samples of germ cells and integrated using Seurat's anchor method alongside nontransplanted samples to provide a single set of UMAP coordinates. All 3 transplant treatments produced distributions of rat germ cells in all cell types, indicating that all stages of spermatogenesis were present in transplanted colonies (Figs 5A and 6A). However, some differences were observed between transplanted rat cell transcriptomes as compared with cells in their native environment. Significantly perturbed pathways were concentrated largely in premeiotic stages, including changes in oxidative phosphorylation, oxidative stress response, and DNA damage responses (S6A Fig).

Following transplantation, SSCs originally from fresh, short-frozen, and long-frozen treatments form a distinct cluster in PCA from fresh and frozen samples (Fig 5B), and this pattern is repeated for each cell type (S7 Fig). Each treatment consisted of 3 or more independent biological replicates. Interestingly, despite the fact that long-frozen stem cells showed little variation from short-frozen following thawing (see Fig 4), after transplantation, gene expression differed more dramatically between these freezing conditions, including a number of genes

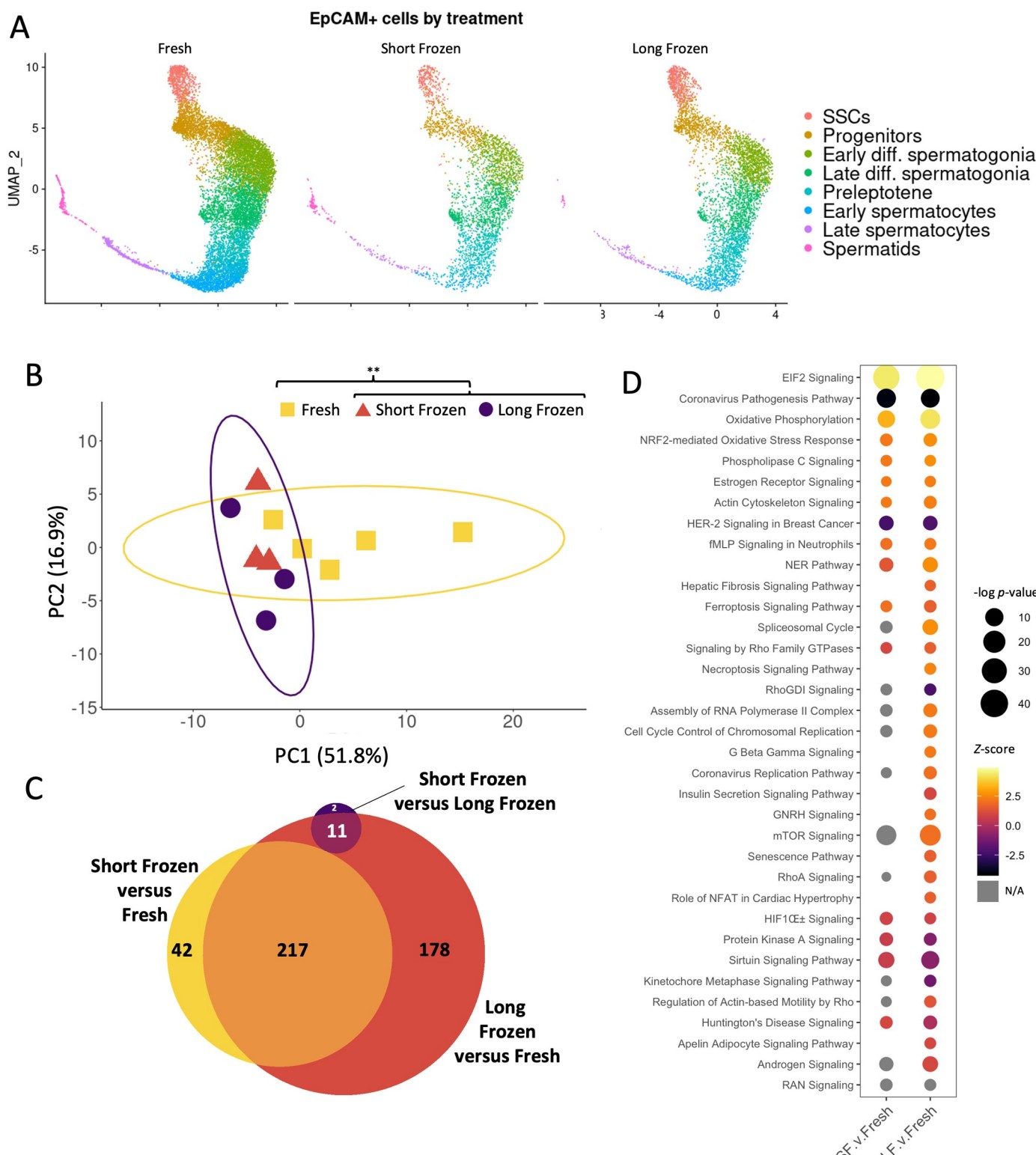

**Fig 4. Long-frozen germ cells show similar, but more perturbed, gene expression profiles to short-frozen following thawing. (A)** Cluster assignments of germ cells divided by treatment reveals presence of all cell types in each treatment. **(B)** PCA plot of pseudobulked gene expression data from the SSC cluster. 95% confidence interval ellipses are shown, counting both short- and long-frozen as a single group. Significance assessed by PERMANOVA, 1,000 permutations, pooling short- and long-frozen as a single group, ** $p < 0.01$. **(C)** Euler diagram of gene expression difference with minimum 1.5-fold change and false discovery rate <0.05 cutoffs applied. **(D)** Ingenuity pathway analysis of DEGs from short-frozen versus fresh and long-frozen versus fresh gene lists. Results ordered by *Z*-

score, showing only results below $p < 0.05$ threshold. All underlying data deposited in NCBI GEO repository (GSE182438). DEG, differentially expressed gene; PCA, principal component analysis; SSC, spermatogonial stem cell; UMAP, uniform manifold approximation and projection.

involved with stem cell self-renewal (Fig 5C, S2 Data). In each of these cases, long-frozen showed higher levels than the transplanted fresh control, and short-frozen samples showed intermediate averages. Cell type–specific pathway analysis of DEGs from long-frozen compared with short-frozen (S3 Data) showed that most dramatic changes in the SSC population, including up-regulation of classic SSC self-renewal pathways such as GDNF family ligand–receptor interactions, MAPK signaling, and cancer signaling (Fig 5D).

This prompted the question of whether long-frozen regenerated tubules would show perturbed proportions of germ cells in each differentiation stage as suggested by UMAP plots of unselected germ cells (Fig 6A). The proportion of long-frozen undifferentiated cells after transplantation was significantly higher than any other treatment (ANOVA, $p < 0.05$). Similarly, transplanted long-frozen differentiating spermatogonia were significantly more abundant than in the fresh rat. Conversely, transplanted long-frozen elongating spermatids were significantly less abundant than in the fresh rat testis. Short-frozen and fresh showed similar proportions for every cell type following transplantation (Fig 6B). While significantly different proportions of cells were observed in certain cell types between treatments, no significant difference in proportion of cells displaying apoptotic markers was observed between treatments (S6B and S6C Fig).

Two of the most significant differently expressed genes when comparing long-frozen to any other treatment were *Prm1* and *Tnp1* (Fig 6C). These genes showed highly significant differences in round spermatids, and *Prm1* was expressed 2.0- and 2.3-fold higher in transplanted fresh or short-frozen, respectively, and 2.8-fold higher in fresh rat as compared with transplanted long-frozen. Similarly, *Tnp1* was expressed 1.7-fold higher in transplanted short-frozen, 2.1-fold higher in transplanted fresh, and 2.3-fold higher in fresh than in transplanted long-frozen. In elongating spermatids, *Prm1* and *Tnp1* did not show any significant differences that passed a $\log_2$-fold change cutoff of ±0.585. Other chromatin proteins also showed some significant changes. *Hils1* and *H1fnt* are both spermatid-specific histone subunits and also showed significant down-regulation in round spermatids in the transplanted long-frozen as compared with the transplanted fresh cells, but not in elongating spermatids. For *Tnp2*, a significantly higher value of 1.6-fold was observed in fresh round spermatids compared with transplanted long-frozen and 1.4-fold higher in transplanted short-frozen. However, no difference was observed in *Prm2* in round spermatids of any of the treatments.

Following transplantation, long-frozen samples showed reduced capability to fully differentiate, displaying a significantly enriched stem cell population coupled with enhanced stem cell signaling. Long-frozen round spermatids expressed less of key chromatin remodeling genes, followed by a significant loss of elongating spermatids as compared with short-frozen samples.

## Discussion

The rat has long been a vital model for male reproduction [1]. We show here for the first time an scRNA-seq analysis of transcriptomic changes in rat spermatogenesis. EpCAM selection provided us with a clearer picture of the early stages of spermatogenesis, as it allowed for identification of cell types that were extremely rare in the unselected rat testicular samples, such as stem cells and progenitor cells. Rat germ cells were classified into clusters matching the developmental changes in mammalian spermatogenesis [13,14,23,24]. The SSC cluster shows up-regulated pathways including GDNF ligand receptor interactions, consistent with GDNF

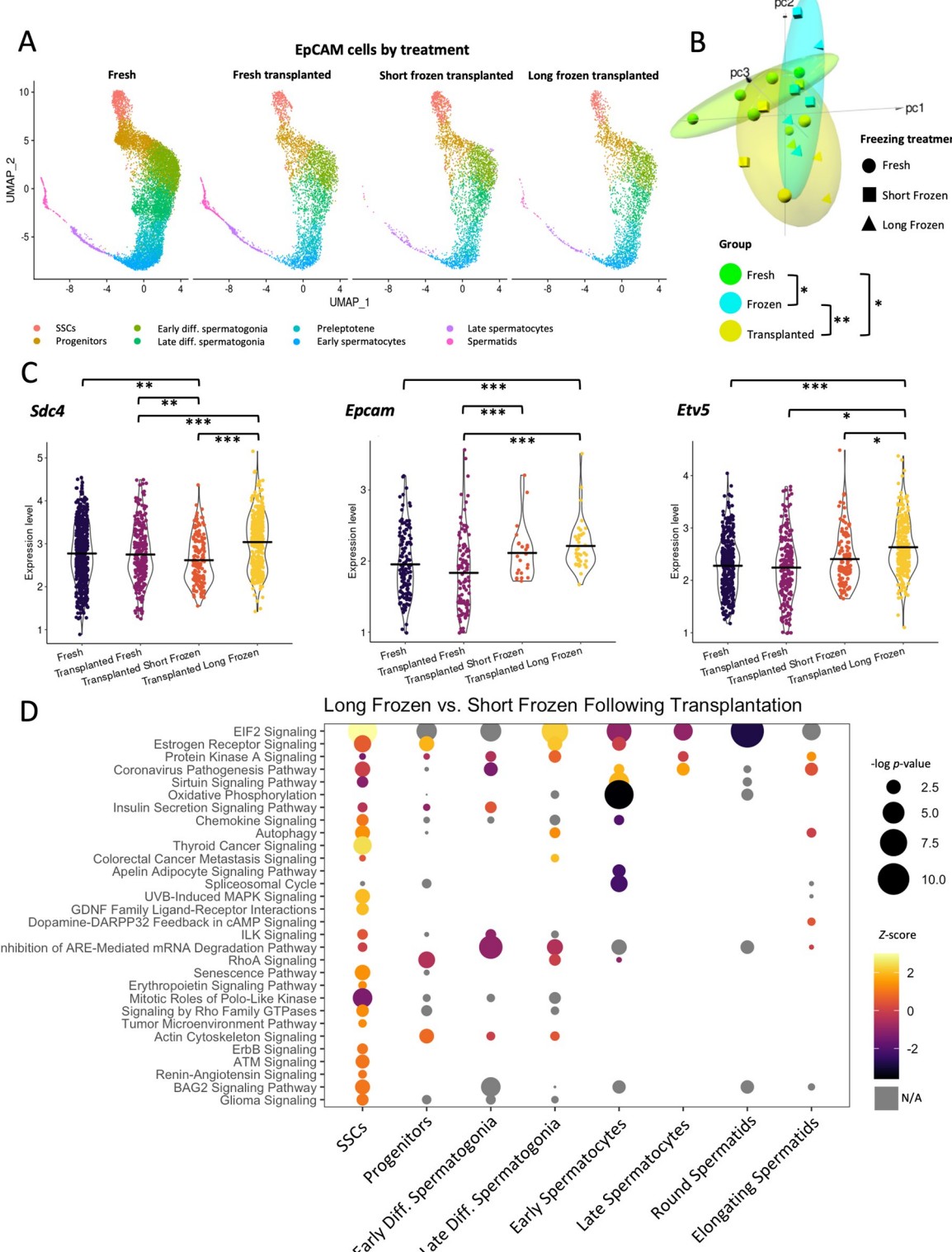

**Fig 5. Transplanted colonies show enhanced activity of SSC genes within the stem cell compartment following long freezing treatment. (A)** Distribution of EpCAM-selected cells from transplanted testes, split by treatments showing all cell type assignments. **(B)** PCA plots for SSCs generated from pseudobulk data for each replicate. 75% confidence intervals are shown for fresh, frozen, and transplanted groups. Significance assessed by PERMANOVA (1,000 permutations, treating short- and long-frozen thawed samples as one group and all transplanted samples as another, $^*$ $p < 0.05$, $^{**}$ $p < 0.01$) **(C)** Expression by treatment for 3 genes that mark undifferentiated

spermatogonia. Each cell with an expression level greater than zero is shown. Significance values were assayed on all cells by Wilcoxon rank sum test via Seurat's FindMarkers function. All underlying data deposited in NCBI GEO repository (GSE182438). **(D)** Ingenuity pathway data generated from DEGs between transplanted short-frozen and transplanted long-frozen clusters. High *Z*-score indicates up-regulation of the pathway in long-frozen cells relative to short-frozen. Nonsignificant *Z*-scores have been excluded. DEG, differentially expressed gene; PCA, principal component analysis; SSC, spermatogonial stem cell; UMAP, uniform manifold approximation and projection.

being indispensable for SSC self-renewal [34–36], along with JAK/STAT [37], EGF [38,39], and NANOG [40] signaling.

Following thawing, we focused on the stem cell cluster, because in any therapeutic application, these cells would form the basis of regenerated tissue. We saw transcriptomic differences in both short-frozen and long-frozen SSC populations as compared with fresh cells, but few gene expression differences between short- and long-frozen cells. The expression changes were consistent with cell damage and shock from freezing. A number of studies have looked at single-cell transcriptomic effects of freezing on a variety of cell lines and tissues, with a range of outcomes from minimal or nondetectable transcriptomic changes [41,42] to up-regulation of heat shock proteins [43] and stress signatures [44]. In all cases, these studies revealed little difference in cell type clustering between thawed samples and fresh samples. In this study, long-frozen cells showed similar changes to short-frozen in direction but more extreme in magnitude of expression, and there were comparatively few genes in the short-frozen versus fresh comparison that were not also present in the long-frozen versus fresh. This is evidenced by a larger number of gene expression differences between long-frozen and fresh than short-frozen and fresh. These 2 comparisons showed similar pathway differences but again more pathway perturbation in the long-frozen versus fresh comparison than short-frozen versus fresh. Among the top hits in the pathway analysis for both treatments were new translation, oxidative phosphorylation, and mitochondrial dysfunction, consistent with stress response due to freezing. Some caution is warranted as some of these differences are also observed in the transition between stem cells and progenitors [45]. Following freezing in liquid nitrogen, SSCs show a variety of stress-induced transcriptional changes compared with fresh cells, changes that are more pronounced in long-frozen samples than short-frozen.

Upon transplantation, testicular cells from all 3 treatment groups were able to successfully colonize recipient mouse testes. Long-frozen cells showed dramatically lower amounts of colony formation, indicating either fewer SSCs per $10^5$ cells that survive to colonize or cells have lost colonization capacity following long freezing treatment. *LacZ*-stained colonies from all 3 treatments were able to generate full spermatogenesis, including spermatozoa. However, from histological analysis, significantly fewer tubules with visible sperm were observed following transplantation of long-frozen samples. This is consistent with assessment of cell numbers in the scRNA-seq data, where long-frozen samples showed significantly more undifferentiated cells and significantly less late spermatids, despite equivalent time in vivo to produce differentiating cells. This indicates that the long freezing treatment has had a detrimental effect on the cells' ability to regenerate tissue that is perpetuated to cell lineages originating from the originally transplanted cells. In order to account for any potential changes in the genetic background of the rats maintained in our laboratory over the 20-year period while cells were frozen, we performed a comparison of genomic alignments looking for mutations relative to the rat reference. No systematic difference correlating with treatment was observed in SNPs or indels, and transplantation efficiency and histology of fresh rats in the present study were comparable to those performed with the same line of rats approximately 20 years prior to this study [46–48]. Together, these indicate that the observed differences between fresh and long-

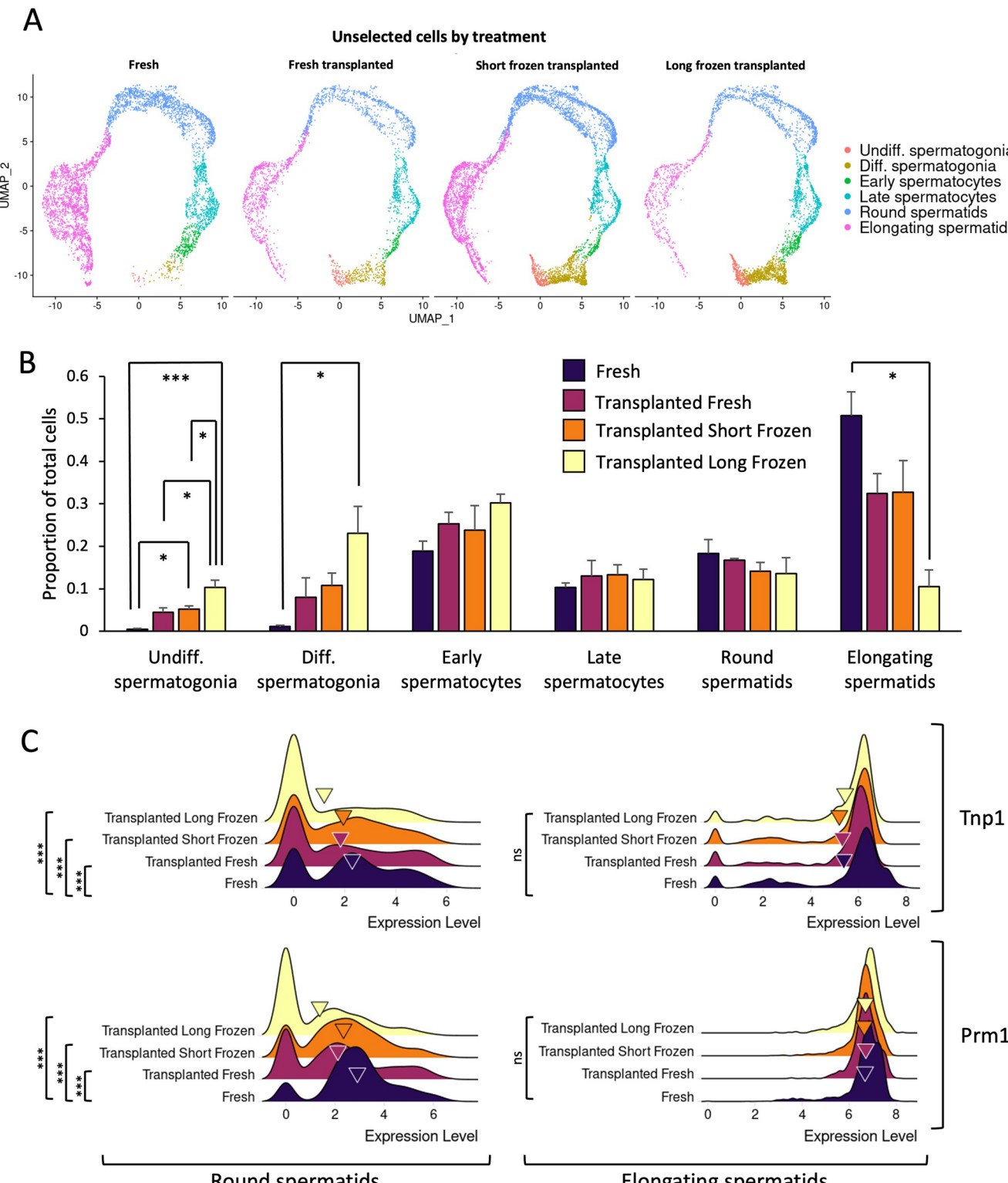

**Fig 6. Transplanted long-frozen cells show fewer terminally differentiated cells. (A)** Distribution of transplanted unselected cells between treatments showing all cell type assignments. All underlying data deposited in NCBI GEO repository (GSE182438). **(B)** Mean number of cells per cell type in the unselected samples by treatment. Error bars denote SEM. Significance was assessed by ANOVA performed separately on each cell type. * $p < 0.05$, ** $p < 0.01$, *** $p < 0.001$ Individual observations recorded in S4 Data. **(C)** Ridge plots of *Prm1* and *Tnp1* expression in round spermatids and elongating spermatids. Significance assessed by FindMarkers function in Seurat. Significant values are shown only where there is a minimum of 1.5-fold change in gene

expression between treatments. Triangles indicate mean expression values. All underlying data deposited in NCBI GEO repository (GSE182438). UMAP, uniform manifold approximation and projection.

frozen were not due to changes in the stock rats over time. Reagents for cryopreservation media were matched to those used in the 1990s in our laboratory for the long-frozen samples, but differences in batches represents an unavoidable possible variable in an experiment spanning over 20 years.

Following transplantation, the transcriptomes of all 3 transplanted treatments clustered in PCA distinctly from the corresponding fresh and newly thawed samples, indicating that transcriptional differences occurred in rat germ cells hosted in a mouse somatic environment. However, long-frozen SSCs showed distinct differences to fresh and short-frozen cells following transplantation. Notably, short- and long-frozen cells showed larger gene expression differences following transplantation than immediately after thawing, indicating that differences between short- and long-frozen cells became more pronounced, not less, after cells grew and divided in a host. Given that these cells have been dividing in vivo for considerable time following thawing, we speculate that the differences may be due to genetic or epigenetic changes to be inherited on the timeline of months. The higher values of key stem cell genes and pathways in the long-frozen suggests higher levels of stem cell self-renewal pathways. Unselected cell counts indicate proportionately and significantly more undifferentiated spermatogonia in the long-frozen treatment group. All transplanted samples showed higher percentages of stem cells than in the rat testis but long-frozen significantly higher than the other treatments. We can attribute the higher percentage in all transplanted samples as compared with fresh to an impairment of normal rat differentiation in the mouse host, but long-frozen samples showed greater impairment, suggesting a reduced amount of differentiation from the stem cell state. At the other end of the differentiation program, lower numbers of cells show expression of *Prm1* in round spermatids, correlating with a lower number of elongating spermatids. This lack of *Prm1*-expressing cells (and other transcripts involved in histone replacement) cannot be explained by lower numbers of UMIs in long-frozen cells as long-frozen spermatids have higher UMI counts than fresh (S3 Fig) and suggests a partial block in postmeiotic differentiation preventing cells from completing spermatogenesis. Our study shows that in terms of therapeutic applications, long-term storage makes recovering sperm from regenerated tissue more challenging. This indicates an urgent need for further studies to determine the underlying cause so that cells can be screened following thawing because transcriptional cues at that time are insufficient to predict poor spermatogenesis following engraftment.

After cryopreservation for >23 years, rat testes cells were able to colonize niches in recipient mice and produce fully differentiated sperm. Previous studies have shown that after transplantation, sperm produced are able to fertilize eggs [49], even after cryopreservation [12]. In this study, following thawing, short- and long-frozen cells showed similar transcriptional profiles, indicating short-term stress, albeit more pronounced in the long-frozen. Crucially, long-frozen samples show a reduced capacity to establish colonies and a reduced capacity to differentiate, resulting in lower numbers of terminally differentiated cells. The length of freezing can have a dramatic impact on the cells' ability to differentiate, but this is only apparent after the tissue begins to regenerate—any clinical use of cells that looks only at biomarkers following thawing, even those that take length of cryopreservation into account, may miss important differences in the recovery of spermatogenesis. These results underscore the importance of exercising caution when interpreting results of clinical translations of stem cell cryopreservation where samples are cryopreserved for short periods of time.

## Materials and methods

### Study design

Cells from Sprague-Dawley rats were analyzed via scRNA-seq after the following treatments: cells freshly digested, cells frozen for a short period (<4 months), and cells frozen for a long period (>23 years). Cells for each treatment came from the same line of rats maintained continuously over the 24-year period with no addition of outside animals. EpCAM+ cells (to enrich for early spermatogonia) were encapsulated. Then, cells for each of these treatments were transplanted into recipient nude mice, and after at least 3 months, both unselected and EpCAM+ fractions were analyzed via scRNA-seq. The goal was to determine the immediate and persistent changes after short-term freezing contrasted with long-term freezing. Each biological replicate came from a separate animal, and all rats used were age-matched to the long-frozen animals (8 to 10 weeks of age). The following numbers of biological replicates were used: fresh unselected, 4 replicates; fresh EpCAM+, 5 replicates; short-frozen EpCAM+, 3 replicates; long-frozen EpCAM+, 3 replicates; transplanted fresh unselected, 4 replicates; transplanted fresh EpCAM+, 5 replicates; transplanted short-frozen unselected, 5 replicates; transplanted short-frozen EpCAM+, 4 replicates; transplanted long-frozen unselected, 3 replicates; transplanted long-frozen EpCAM+ 3 replicates. Replicates are shown in S4 Fig and where unselected and EpCAM+ samples share the same identifier, these came from the same animal. All long-frozen samples were derived from 3 original independent biological replicates. All animal protocols were approved by University of Pennsylvania Institutional Animal Care and Use Committee (protocol number 800375).

### Tissue isolation

Cells were isolated from the testes of 8- to 10-week-old rats transgenic for the LacZ gene under the metallothionine promoter [46]. Tissue was chopped into fine pieces and incubated in collagenase (Sigma) at a concentration of 1 mg/ml in HBSS (Gibco) for 15 minutes at 37˚C. Cells were spun down for 1 minute at 600$g$, then resuspended in warm Trypsin (Gibco, 0.25%) with 20% DNase solution (Sigma, 7 mg/ml dissolved in HBSS). Tissue was pipetted for 2 minutes with a 10-ml pipette and incubated at 37˚C for 5 minutes. Then, tissue was pipetted for another 2 minutes and incubated at 37˚C for 3 minutes. FBS (Sigma F2442) was added to stop the digestion. Additional DNase was added until no turbidity was visible. Cells were washed in PBS-S twice (PBS [Gibco] with 1% FBS, 10 mM HEPES [Sigma Aldrich], 1 mg/ml glucose [Sigma Aldrich], 1 mM pyruvate [Gibco], 50 units/ml penicillin [Gibco], 50 ug/ml streptomycin [Gibco] prepared as described in [50]). All spins were 5 minutes at 600$g$. Mouse cells were isolated following the same procedure from adult C57 BL6 mice (Jackson Laboratories), with the exception that the collagenase step was skipped.

### Cryopreservation

Cells were frozen in DMEM-C with final concentrations of 10% DMSO (Sigma D2650) and 10% FBS at concentrations between 1 and $4 \times 10^7$ cells/ml. Short-frozen cells were prepared in exactly the same manner as the cryopreservation protocol used in 1996, including the freezing containers. While using the same lots of reagents was impossible, reagents were matched and of the same quality. Cell suspension was resuspended at $8 \times 10^7$ cells/ml in DMEM-C (DMEM with the addition of 2.2 g/L sodium bicarbonate, 100 units/ml penicillin [Life Technologies], and 100 μg/ml streptomycin [Life Technologies], 1.25 g/L sodium bicarbonate, 2 mM glutamine [Life Technologies], 0.4 mM pyruvic acid [P-5280; Sigma], 6 mM lactic acid [L-4263; Sigma], and 0.1 mM 2-mercaptoethanol [M-7522; Sigma]; see [51]). An equal volume of 2X

freezing solution made up of 20% DMSO, 20% FBS, and remainder DMEM-C was prepared and added dropwise to each vial. Vials were immediately place in cardboard cartons filled with tissue paper and moved to the −80°C freezer. After 24 hours, the vials were transferred to liquid nitrogen for storage. "Long-frozen" samples were stored in liquid nitrogen for 23 to 24 years. "Short-frozen" samples were stored for at least 1 month in liquid nitrogen and not more than 4 months.

## Thawing

To thaw the samples, samples were thawed in a 37°C water bath and then immediately after becoming liquid a solution of DMEM-C with 200 mM trehalose [T9449; Sigma] and 5% DMSO was added dropwise to a total volume of 3 ml. An additional 2 ml of DMEM-C with 200 mM trehalose was added, and the cells spun down at 600$g$ at 4°C for 5 minutes. A volume of 1 ml of DNase solution in PBS (7 mg/ml) and 1 ml PBS-S were added, pipetted gently but thoroughly to ensure no clumping and 8 ml of PBS-S was added and cells were counted on a hemocytometer with trypan blue (1:1). Typical viability was between 5% to 15% for thawed samples.

## Single-cell RNA-seq and bioinformatics

As the viability postthaw of adult rat cells was only 5% to 10%, comparable to mouse germ cells after long-term cryopreservation [12], we used an antibody-based kit to remove dead cells prior to RNA-seq. Following isolation and thawing as appropriate, dead cells were removed using the Miltenyi Dead Cell Removal Kit. Approximately 100 ul of beads was used per $10^7$ cells and incubated at room temperature for 15 minutes. Cells were diluted with 1 ml of binding buffer including 3.5 mg/ml DNase. Dead cells were bound using MACS MS columns (Miltenyi) and rinsed 4 times with 0.5 ml of binding buffer. The collected cells resulted in a viability of around 80% for frozen samples. For samples selected for EpCAM, cells were incubated with mouse anti-rat EpCAM antibody (clone GZ1 produced by Dr. Gottfried Dohr in the Medical University of Graz, Austria [52]) for 20 minutes at 4°C. Samples were washed twice in PBS-S, resuspended and incubated with secondary antibody conjugated to anti-mouse magnetic microbeads (Miltenyi) for 20 minutes at 4°C. Cells were washed twice and selected via MACS MS columns (Miltenyi). In order to encapsulate, a minimum viability of 80% was required to include a sample, but typical viability was 90% to 95%.

For fresh and transplanted samples, in addition to EpCAM+ cells prepared as above, unselected cells were prepared from the same samples by keeping aside a fraction of the cells without EpCAM selection.

Cells were encapsulated and libraries generated using the inDrop system (1CellBio) per manufacturer's protocol. Each biological replicate was encapsulated individually, but library preps were generated in batches of 3 treatments (e.g., fresh, short-frozen, long-frozen or transplanted fresh, transplanted short-frozen, and transplanted long-frozen). Libraries were sequenced on a NextSeq500 sequencer (Illumina) using a 75-cycle high-output sequencing kit to a minimum depth of 30k reads per cell. Data were processed using the indrops.py pipeline provided by 1CellBio (https://github.com/indrops/indrops), which uses Bowtie (version 1.3.0) [53] to align reads to the rat genome, in this case the Rattus norvegicus 6.0 DNA primary assembly along with the corresponding GTF file (v6.0.85) from Ensembl [54]. Gene counts were analyzed with Seurat v3.1 [55] for clustering, integration, and differential gene expression and Monocle version 3 [56] for pseudotime. Ingenuity Pathway Analysis (QIAGEN; https://www.qiagenbioinformatics.com/products/ingenuity-pathway-analysis) was used for all

pathway analyses. In all cases, gene lists were used with a minimum fold change cutoff of ±1.5 and *p*-adjusted value (*pAdj*) value of ≤0.05.

For transplanted samples, cells were identified as mouse or rat by aligning all cells to both mouse and rat transcriptomes. Each cell was scored as "mouse" or "rat" on the basis of which successfully aligned more reads, or "unknown" if neither mouse nor rat reads were more than 5% higher than the other. To confirm this protocol worked, this was also tested on samples generated from pure rat and pure mouse with greater than 99% efficiency.

To generate principal components, the gene expression for each replicate was averaged (mean) across all cells within each assigned cell type cluster. These pseudobulked data were used to generate distance matrices and plot PCs between treatment groups.

For analysis of mutations, BAM files generated from the indrops.py pipeline were used as the basis of GATK's RNAseq short variant discovery (SNPs + Indels) [57]. BAM files were labeled with picard [58], indexed with samtools [59], corrected for splicing and halplotypes called via GATK. PCA was generated via SNPRelate R package [60].

## Transplantation

Cells were prepared either as a fresh isolation or thawed cells. All samples were enriched for live cells and EpCAM-selected as described above. Transplants generated for single-cell analysis were transplanted at the highest practical cell density ($5 \times 10^7$ cells/ml) in order to maximize colony formulation, with at least 3 biological replicates per treatment. Additional transplants were performed at $3 \times 10^6$ for colony counting (for long-frozen samples, due to the low stem cell number, as many cells as possible were injected and the total number injected used to calculate stem cell number). Transplantation procedure was performed as described previously [61]. After between 10 weeks and 4 months, animals were killed according to institutional guidelines. Testes were extracted, weighed, and the tunica removed. For encapsulation, cells were prepared as described above. For colony counting, transplanted testes were stained with X-gal as described previously [62].

## Supporting information

**S1 Fig. Unbiased clustering of all cells, including contaminating mouse cells. (A)** Transplantation colony counts using unselected fresh rat cells are shown compared with those from studies performed approximately 20 years earlier from the same rat line and laboratory [47,48]. Novel observations shown in S4 Data. **(B)** UMAP projection of unbiased clustering of all unselected cells. **(C)** Unselected cells colored by species of origin. **(D)** Identity assignments of unselected rat and mouse cells in transplanted testes by alignment to both transcriptomes. Cells were assigned an identity to whichever species produced a higher number of UMI hits above a 5% threshold. Each sample represents an independent biological replicate (S4 Data). **(E)** UMAP projection of unbiased clustering of all EpCAM+ cells. **(F)** EpCAM+ cells colored by species of origin. **(G)** Identity assignments of EpCAM rat and mouse cells in the same manner as unselected (S4 Data). Cells with mouse or unknown identity assignments were removed from the analysis. For all UMAPs, underlying data deposited in NCBI GEO repository (GSE182438). UMAP, uniform manifold approximation and projection; UMI, unique molecular identifier. (DOCX)

**S2 Fig. Removal of somatic cells. (A)** Key somatic genes that identify clusters 4, 18, 24, and 29 as somatic clusters. **(B)** Key somatic genes that identify clusters 9, 12, 16, and 22 as somatic clusters. All somatic clusters were removed, and cells were reclustered. All underlying data deposited in NCBI GEO repository (GSE182438). UMAP, uniform manifold approximation

and projection.
(DOCX)

**S3 Fig. Quality control metrics of samples following filtering.** Genes/cell indicates the number of distinct genes that have one or more transcripts per cell. UMI/cell indicates the unique molecular identifier count for each cell. % mitochondrial reads shows the relative percentage of mitochondrial reads to chromosomal reads per cell. **(A)** All unselected cells, split by treatment. **(B)** All EpCAM+ cells, split by treatment. **(C)** Unselected cells grouped by cell type and colored by treatment. **(D)** EpCAM+ cells grouped by cell type and colored by treatment. All underlying data deposited in NCBI GEO repository (GSE182438). Fr, fresh; LF, long-frozen; SSC, spermatogonial stem cell; SF, short-frozen; T-Fr, transplanted fresh; T-LF, transplanted long-frozen; T-SF, transplanted short-frozen; UMAP, uniform manifold approximation and projection; UMI, unique molecular identifier.
(DOCX)

**S4 Fig. Expression patterns of key germ cell genes. (A)** UMAP projections of 9 germ cell marker genes in unselected germ cells. **(B)** UMAP projections of 9 germ cell marker genes in EpCAM+ germ cells. All underlying data deposited in NCBI GEO repository (GSE182438). UMAP, uniform manifold approximation and projection.
(DOCX)

**S5 Fig. Clustering of biological replicates. (A)** All unselected cells, split by treatment and colored by replicate. **(B)** All EpCAM+ cells, split by treatment and colored by replicate. TLF3 and TLF4 are the same biological replicate but encapsulated on different days, otherwise each designation is a different biological replicate. **(C)** PCA plot derived from SNP/short indel data derived from mRNA alignments using the GATK mRNA pipeline, using the same data as Fig 4B. Percentage of variation explained by each principal component indicated on the axes. No significant difference (ns) was detected via PERMANOVA. All underlying data deposited in NCBI GEO repository (GSE182438). Fr, fresh; LF, long-frozen; PCA, principal component analysis; SF, short-frozen; SNP, single nucleotide polymorphism; T-Fr, transplanted fresh; T-LF, transplanted long-frozen; T-SF, transplanted short-frozen; UMAP, uniform manifold approximation and projection.
(DOCX)

**S6 Fig. Analysis of the effect of transplantation on spermatogenesis. (A)** Ingenuity pathway data generated from DEGs between fresh rat cells and transplanted fresh rat for each cell type. High $Z$-score indicates up-regulation of the pathway in fresh cells relative to transplanted. Nonsignificant $Z$-scores have been excluded (S4 Data). **(B)** An apoptosis score was generated for each cell. This was done by taking a list of proapoptotic genes (derived from Ingenuity's database and listed below) and used Seurat's AddModuleScore function to produce an apoptosis score for each cell. These scores are shown in the UMAP presentations (left). In addition, any cell with an apoptosis score over 0.1 (arbitrary cutoff) was designated as apoptotic, and the fraction of apoptotic cells per replicate for each cell type was calculated and the mean fraction is shown (right, S4 Data). Error bars designate SEM. **(C)** Above process was repeated for EpCAM+ cells. All underlying data deposited in NCBI GEO repository (GSE182438). Proapoptotic gene list: *Acin1*, *Apaf1*, *Bad*, *Bak1*, *Bax*, *Bcl2l11*, *Bcl2l14*, *Bid*, *Bik*, *Bmf*, *Bnip3l*, *Bok*, *Casp2*, *Casp3*, *Casp6*, *Casp7*, *Casp8*, *Casp9*, *Casp12*, *Dapk1*, *Dapk2*, *Dapk3*, *Dedd*, *Dffa*, *Diablo*, *Ercc2*, *Ercc3*, *Fas*, *Faslg*, *Foxo3*, *Tnf*, *Tnfrsf10b*, *Tnfrsf1a*, *Tnfrsf1b*, *Tnfsf14*, *Tp53*, *Tradd*, *Traf3*. DEG, differentially expressed gene; LF, long-frozen; SF, short-frozen; SSC, spermatogonial stem cell; UMAP, uniform manifold approximation and projection.
(DOCX)

**S7 Fig. PCA plots of all replicates and treatments by cell type.** For each cell type, pseudo-bulked log-normalized gene expression data were used to generate principal components. 75% confidence intervals are projected for fresh samples, short-, and long-frozen grouped together and all transplanted cells as a single group. All underlying data deposited in NCBI GEO repository (GSE182438). PCA, principal component analysis; SSC, spermatogonial stem cell. (DOCX)

**S1 Data. Cell-specific markers.**
(XLSX)

**S2 Data. Comparisons between treatments after transplantation all cells.**
(XLSX)

**S3 Data. Comparison of long- and short-frozen by cell type.**
(XLSX)

**S4 Data. Underlying data for figures.**
(XLSX)

**S1 Code. R script containing code used to analyze all single-cell RNA-seq data.**
(R)

## Acknowledgments

We thank Nilam Sinha, Kotaro Sasaki, Xin Wu, Keren Cheng, and Jeremy Wang for helpful advice. We also thank C. Freeman, R. Naroznowski, and D. Lee for animal maintenance.

**Lead author statement**

Further information and requests for resources and reagents should be directed to and will be fulfilled by the lead author, Eoin Whelan (ewhelan@vet.upenn.edu).

**Materials availability**

This study did not generate new unique reagents.

## Author Contributions

**Conceptualization:** Eoin C. Whelan, Ralph L. Brinster.

**Data curation:** Eoin C. Whelan.

**Formal analysis:** Eoin C. Whelan.

**Funding acquisition:** Ralph L. Brinster.

**Investigation:** Eoin C. Whelan, Fan Yang, Mary R. Avarbock, Megan C. Sullivan.

**Methodology:** Eoin C. Whelan, Fan Yang, Daniel P. Beiting, Ralph L. Brinster.

**Project administration:** Eoin C. Whelan, Ralph L. Brinster.

**Supervision:** Ralph L. Brinster.

**Visualization:** Eoin C. Whelan.

**Writing – original draft:** Eoin C. Whelan.

**Writing – review & editing:** Eoin C. Whelan, Fan Yang, Daniel P. Beiting, Ralph L. Brinster.

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
