## [Editor Report · Decision Letter 0]

16 Dec 2021

Dear Dr Whelan, 

Thank you for submitting your revised manuscript entitled "Long-term Cryopreservation of Rat Spermatogonial Stem Cells Causes Defects in Regenerated Spermatogenesis" for consideration as a Research Article by PLOS Biology. I have now discussed your manuscript with the Academic Editor, who is satisfied by the changes made in response to his/her previous concerns, and we would therefore like to send your submission out for external peer review. As a note, while the Academic Editor is satisfied by the new analyses, we will also need the reviewers to be convinced that the conclusions are strongly supported. 

Before we can send your manuscript to reviewers, we need you to complete your submission by providing the metadata that is required for full assessment. To this end, please login to Editorial Manager where you will find the paper in the 'Submissions Needing Revisions' folder on your homepage. Please click 'Revise Submission' from the Action Links and complete all additional questions in the submission questionnaire.

Once your full submission is complete, your paper will undergo a series of checks in preparation for peer review. Once your manuscript has passed the checks it will be sent out for review. To provide the metadata for your submission, please Login to Editorial Manager (https://www.editorialmanager.com/pbiology) within two working days, i.e. by Dec 20 2021 11:59PM.

If your manuscript has been previously reviewed at another journal, PLOS Biology is willing to work with those reviews in order to avoid re-starting the process. Submission of the previous reviews is entirely optional and our ability to use them effectively will depend on the willingness of the previous journal to confirm the content of the reports and share the reviewer identities. Please note that we reserve the right to invite additional reviewers if we consider that additional/independent reviewers are needed, although we aim to avoid this as far as possible. In our experience, working with previous reviews does save time. 

If you would like to send previous reviewer reports to us, please email me at lsmith@plos.org to let me know, including the name of the previous journal and the manuscript ID the study was given, as well as attaching a point-by-point response to reviewers that details how you have or plan to address the reviewers' concerns. 

Given the disruptions resulting from the ongoing COVID-19 pandemic and the upcoming holidays, please expect some delays in the editorial process. The PLOS Biology office will be closed from Dec 22-31st, and while I will do my best to start inviting reviewers before the holiday, I may not be able to secure a complete set before the break. We apologize in advance for any inconvenience caused and will do our best to minimize impact as far as possible.

Kind regards,

Lucas

Lucas Smith

Associate Editor

PLOS Biology

lsmith@plos.org

---

## [Decision Letter · Decision Letter 1]

15 Feb 2022

Dear Dr Whelan,

Thank you for submitting your manuscript entitled "Long-term Cryopreservation of Rat Spermatogonial Stem Cells Causes Defects in Regenerated Spermatogenesis" for consideration as a Research Article at PLOS Biology. Thank you also for your patience as we completed our editorial process, and please accept my apologies for the delay in providing you with our decision. Your manuscript has been evaluated by the PLOS Biology editors, an Academic Editor with relevant expertise, and by two independent reviewers.

As you will see, the reviewers find your study interesting and worth pursuing for publication, but they also raise several concerns that need to be addressed to improve the manuscript and to clarify some points. The reviewers think that you should consider the potential effects of other factors on the long-term cryopreservation, thus we would like you to discuss in the text these limitations of the study. Reviewer 1 thinks you should test if the SSCs can differentiate into sperm even after long-term storage and that the title should also reflect this. Reviewer 2 also raises several points that you should address.

In light of the reviews (attached below), we are pleased to offer you the opportunity to address the comments from the reviewers in a revised version that we anticipate should not take you very long. We will then assess your revised manuscript and your response to the reviewers' comments and we may consult the reviewers again.

We expect to receive your revised manuscript within 1 month.

**IMPORTANT - SUBMITTING YOUR REVISION**

3. Resubmission Checklist

a) *PLOS Data Policy*

b) *Published Peer Review*

Sincerely,

Ines

--

Ines Alvarez-Garcia, PhD

Senior Editor

PLOS Biology

on behalf of

Lucas Smith

Associate Editor

PLOS Biology

lsmith@plos.org

Reviewers' comments

Rev. 1:

In this paper, Brinster and colleagues transplanted long-cryopreserved rat SSCs into nude mice to find colonization of those SSCs and the production of sperm. The cryopreservation period was amazingly over 23 years. This length of time is realistic and practical when this procedure will become a clinical procedure for human patients. Thus, it is impressive and informative.

Authors, however, rather focused on a negative side of the results that such a long-preservation would affect spermatogenic competence of the SSCs, leading to defective spermatogenesis. They resorted to scRNA-seq analysis and demonstrated that long frozen sample was distinct from those of fresh and short frozen. Interestingly, the scRNA seq of transplanted samples showed that the transplanted germ cells tended to remain as SSC or spermatogonia, even when fresh sample was transplanted. This may suggest that the rat SSCs in the mouse testis had some difficulty to differentiate. It is therefore interesting to test if this would happen also when mouse SSCs were transplanted to mouse testis.

I'm not fully convinced that long-term cryopreservation, but not short-term, causes spermatogenic defects. Thus, if I'm an author of this study, I would like to change the title even though most of the text remains as it is. The title I prefer is something like these.

Revival of Spermatogenesis after over 20 years of Cryopreservation of Rat Spermatogonial Stem Cells

Resurgence of spermatogenesis after more than 20 years of cryopreservation of rat spermatogonial stem cells

Major points:

1. It is yet difficult for me to accept the conclusion of this paper. There could be several parameters other than long-term cryopreservation that influence the result of this study. I suspect that the quality of cryoprotectant could be critical. Quality of such reagents might have affected the result of this study.

2. It is not clear how many times the cryopreservation was performed in days of 23 years ago. In other words, how many rats were used for the cryopreservation of long frozen sample?

3. In the results, it is written that stocks of Norwegian rat testis cells that were cryopreserved over 23 years ago was used in this study. On the other hand, in the materials and methods, it is written that cells from Sprague-Dawley rats were analyzed. Does this mean that different strains of rat were used in this study?

Minor points:

3. In Fig. 6B, four colors of each, light yellow, orange, dark red, and dark blue, seems to show fresh, transplanted fresh, transplanted short frozen, and transplanted long frozen, respectively. On the other hand, in 6C, the samples and the colors match in a different way. This is really a minor issue but a bit confusing.

Rev. 2: Sue Hammoud – note that this reviewer has signed her review

This study by Whelan et. al. examines the molecular and functional effects of extended periods of cryopreservation on the ability of rat spermatogonial stem cells to successfully regenerate spermatogenesis. To explore this question, the authors perform a comparative scRNA-seq analysis of isolated fresh germ cell populations vs. germ cell populations isolated after transplantation. This elegant work revealed two key insights about SSC fertility preservation: First, long term crypreservation of SSCs have reduced stem cell transplantation efficiency, reduced stem cell differentiation capacity, as well as increased disorganized tubules in areas where germ cell development is restored. Second, the authors show significant molecular changes in transcriptome of fresh vs. frozen germ cells. This work is exciting and is of significant value to the reproductive biology and medicine community. Some minor concerns were noted:

1) What is the absolute frequency of tubule disorganization when comparing fresh and frozen samples? How do you distinguish between a disorganized tubule vs. sectioning vs. embedding artifact?

2) Can differences in stem cell activity be due to : changes in cryopreservation media composition over time. Can the differences in regenerative ability of short vs. long term frozen samples (Figure 1C) be due to certain components? Is the age of SSCs matched in fresh vs. cryopreserved samples?

3) How does the Rat germ cell differentiation program compare to rat germ cell differentiation occurring in mouse testis? Although Rat spermatogenesis can occur in the mouse testis, no one really has explored molecular genetic differences in the differentiation program. The data is currently available in this manuscript, and only needs to be mined. Adding fresh sample data to figure 5 will be an interesting comparison?

4) In Figure 6B, is it possible that the labels for the yellow and violet are swapped? The conclusion that longterm frozen cells have a higher fraction of undifferentiated cells and lower percentage of differentiated germ cells is not consistent with the data. The columns may be mislabeled.

5) Is the decrease in spermatid generation in long-term cryopreservation due to block in differentiation or increase post meiotic germ cell pruning/death?

---

## [Editor Report · Decision Letter 2]

14 Mar 2022

Dear Dr Whelan,

Thank you for submitting your revised Research Article entitled "Resurgence of spermatogenesis after more than 20 years of cryopreservation of rat spermatogonial stem cells" for publication in PLOS Biology. I have now obtained advice from the Academic Editor and discussed the revision with the rest of the team.

Based on the discussions, we will probably accept this manuscript for publication, provided you satisfactorily address the following data and other policy-related requests.

In addition, we would like to make a suggestion to improve the title - we know that you have changed it as suggested by one of the reviewers, but we do think that it should be more informative for readers:

"Re-establishment of spermatogenesis after more than 20 years of cryopreservation of rat spermatogonial stem cells reveals an important impact in differentiation capacity"

We expect to receive your revised manuscript within two weeks. 

*Published Peer Review History*

*Press*

Sincerely,

Ines

--

Ines Alvarez-Garcia, PhD

Senior Editor

PLOS Biology

FINANCIAL DISCLOSURE:

Please include grant numbers and the URLs of any funder's website.

Fig. 1C, D; Fig. 2A-C; Fig. 3A, C, D; Fig. 4A, B; Fig. 5A, C; Fig. 5A-C; Fig. S1A-G; Fig. S2A, B; Fig. S3A-D; Fig. S4A, B; Fig. S5A-C and Fig. S6B, C

**In addition, you should make the data you have deposited in the NCBI GEO repository (GSE182438) publicly available, before we proceed with Production.

---

## [Editor Report · Decision Letter 3]

4 Apr 2022

Dear Dr Whelan,

On behalf of my colleagues and the Academic Editor, Masahito Ikawa, I am pleased to say that we can in principle accept your Research Article entitled "Re-establishment of spermatogenesis after more than 20 years of cryopreservation of rat spermatogonial stem cells reveals an important impact in differentiation capacity" for publication in PLOS Biology, provided you address any remaining formatting and reporting issues. These will be detailed in an email that will follow this letter and that you will usually receive within 2-3 business days, during which time no action is required from you. Please note that we will not be able to formally accept your manuscript and schedule it for publication until you have completed any requested changes.

PRESS

Sincerely, 

Ines

--

Ines Alvarez-Garcia, PhD 

Senior Editor 

PLOS Biology
